# Relational Proxies: Emergent Relationships as Fine-Grained Discriminators

**Abhra Chaudhuri**[1]     **Massimiliano Mancini**[2]     **Zeynep Akata**[2,3,4]     **Anjan Dutta**[5]*

[1] University of Exeter      [2] University of Tübingen      [3] MPI for Informatics
[4] MPI for Intelligent Systems      [5] University of Surrey

## Abstract

Fine-grained categories that largely share the same set of parts cannot be discriminated based on part information alone, as they mostly differ in the way the local parts relate to the overall global structure of the object. We propose *Relational Proxies*, a novel approach that leverages the relational information between the global and local views of an object for encoding its semantic label. Starting with a rigorous formalization of the notion of distinguishability between fine-grained categories, we prove the necessary and sufficient conditions that a model must satisfy in order to learn the underlying decision boundaries in the fine-grained setting. We design Relational Proxies based on our theoretical findings and evaluate it on seven challenging fine-grained benchmark datasets and achieve state-of-the-art results on all of them, surpassing the performance of all existing works with a margin exceeding 4% in some cases. We also experimentally validate our theory on fine-grained distinguishability and obtain consistent results across multiple benchmarks. Implementation is available at `https://github.com/abhrac/relational-proxies`.

## 1   Introduction

Fine-grained visual categorization (FGVC) primarily requires identifying category-specific, discriminative local attributes [49, 44, 21]. However, the *relationship* of the attributes with the global view of the object is also known to encode semantic information [6, 5]. Such a relationship can be thought of as the way in which local attributes combine to form the overall object. When two categories share a large number of local attributes, this cross-view relational information becomes the only discriminator. To illustrate this in an intuitive example, Figure 1 shows two fine-grained categories of birds, the White-faced Plover (left and top-right) and the Kentish Plover (bottom-right). Along with color and texture information, the two categories share a large number of local features like beak, head, body, tail and wings. Given such constraints of largely overlapping attribute sets, relational information like the distance between the head and the body, or the angular orientation of the legs with respect to the body remain as the only available discriminators. We thus conjecture that the way the global structure (view) of the object arises out of its local parts (views) must be an *emergent* [30] property of the object which is implicitly encoded as the cross-view relationship. However, all existing methods that consider both global and local information, do so in a *relation-agnostic* manner, *i.e.*, without considering cross-view relationships (we formalize relation-agnosticity in Section 3).

We hypothesize that when two categories largely share the same set of local attributes and differ only in the way the attributes combine to generate the global view of the object, relation-agnostic approaches do not capture the full semantic information in an input image. To prove our hypothesis, we develop a rigorous formalization of the notion of distinguishability in the fine-grained setting. Via our theoretical framework, we identify the necessary and sufficient conditions that a learner must satisfy to completely learn a distribution of fine-grained categories. Specifically, we prove

---

*A. Chaudhuri is with the Department of Computer Science at the University of Exeter. M. Mancini and Z. Akata are with the Cluster of Excellence Machine Learning at the University of Tübingen. A. Dutta is with the Institute for People-Centred AI at the University of Surrey.

36th Conference on Neural Information Processing Systems (NeurIPS 2022).

that a learner must harness both view-specific (relation-agnostic) and cross-view (relation-aware) information in an input image. We also prove that it is not possible to design a single encoder that can achieve both of these objectives simultaneously. Based on our theoretical findings, we design a learner that separately computes metric space embeddings for the relation-agnostic and relation-aware components in an input image, through class representative vectors that we call Relational Proxies.

To summarize, we: (1) provide a theoretically rigorous formulation of the FGVC task and formally prove the necessary and sufficient conditions a learner must satisfy for FGVC, (2) introduce a plug-and-play extension on top of conventional CNNs that helps leverage relationships between global and local views of an object in the representation space for obtaining a complete encoding of the fine-grained semantic information in an input image, (3) achieve state-of-the-art results on all benchmark FGVC datasets with significant accuracy gains.

## 2 Related Work

**Fine-grained visual categorization** Prior works have demonstrated the importance of learning localized image features for FGVC [1, 50, 23], with extensions exploiting the relationship between multiple images and between network layers [25]. The high intra-class and low inter-class variations in FGVC datasets can be tackled by designing appropriate inductive biases like normalized object poses [4] or via more data-driven methods like deep metric learning [7]. Analysing part-specific features along with the global context was demonstrated through part detection based on activation regions in CNN feature maps [16, 48] or via context-aware attention pooling [3]. CNNs can also be modified in novel ways for FGVC by incorporating boosting [27], kernel pooling [8], or by randomly masking out a group of correlated channels during training [10]. Vision Transformers [40], with their ability to attend to specific informative image patches, have also shown great promise in FGVC [42, 13, 24]. To the best of our knowledge, we are the first to provide a rigorous theoretical foundation for FGVC and design a cross-view relational metric learning formulation based on the same.

**Relation modelling in deep learning** Modelling relationships between entities has proven to be a useful approach in many areas of deep learning including deep reinforcement learning [47], object detection [15], question answering [35], graph representation learning [2], few-shot learning [37] and knowledge distillation [31]. The usefulness of modelling relationships between different views of the same image has been demonstrated in the self-supervised context by [33]. All the above works either leverage or aim to learn relationships between entities, the nature of which is assumed to be known *apriori*. Our work breaks free from such assumptions by modelling cross-view relationships as learnable representations that optimize the end-task of FGVC.

**Proxy-based deep metric learning** Motivated by the fact that pairwise losses for deep metric learning incur a significant computational overhead leading to slow convergence, the idea of using proxies for learning metric spaces was first proposed in [28] and enhanced in [38]. Proxies can also be used to emulate properties of pairwise losses by capturing data-to-data relations (instead of just data-to-proxy) leveraging relative hardness of datapoints [18], by making data representations follow the semantic hierarchy inherent in real-world classes [45], or by regularizing sample distributions around proxies to follow a non-isotropic distribution [34]. However, all the above works perform proxy-based metric learning directly on data representations. In contrast, our approach is designed to learn class proxies that can be used not only to capture isolated, view specific (local/global) information for the underlying class, but also to learn the cross-view *relationships* such that they form embeddings in a metric space.

## 3 Relational Proxies

Consider an image $\mathbf{x} \in \mathbb{X}$ with a label $\mathbf{y} \in \mathbb{Y}$. Let $\mathbf{g} = c_g(\mathbf{x})$ and $\mathbb{L} = \{\mathbf{l}_1, \mathbf{l}_2, ... \mathbf{l}_k\} = c_l(\mathbf{x})$ be the global and set of local views of an image $\mathbf{x}$ respectively, where $c_g$ and $c_l$ are cropping functions applied on $\mathbf{x}$ to obtain such views. Let $f$ be an encoder that takes as input $\mathbf{v} \in \{\mathbf{g}\} \cup \mathbb{L}$ and maps it to a latent space representation $\mathbf{z} \in \mathbb{R}^d$, where $d$ is the representation dimensionality. Specifically, the representations of the global view $\mathbf{g}$ and local views $\mathbb{L}$ obtained from $f$ are then denoted by $\mathbf{z}_g = f(\mathbf{g})$ and $\mathbb{Z}_\mathbb{L} = \{f(\mathbf{l}) : \mathbf{l} \in \mathbb{L}\} = \{\mathbf{z}_{l_1}, \mathbf{z}_{l_2}, ... \mathbf{z}_{l_k}\}$ respectively. Let $R : (\mathbf{g}, \mathbb{L}) \to \mathbf{r}$ be a random variable that encodes the relationships $\mathbf{r}$ between the global ($\mathbf{g}$) and the set of local ($\mathbb{L}$) views.

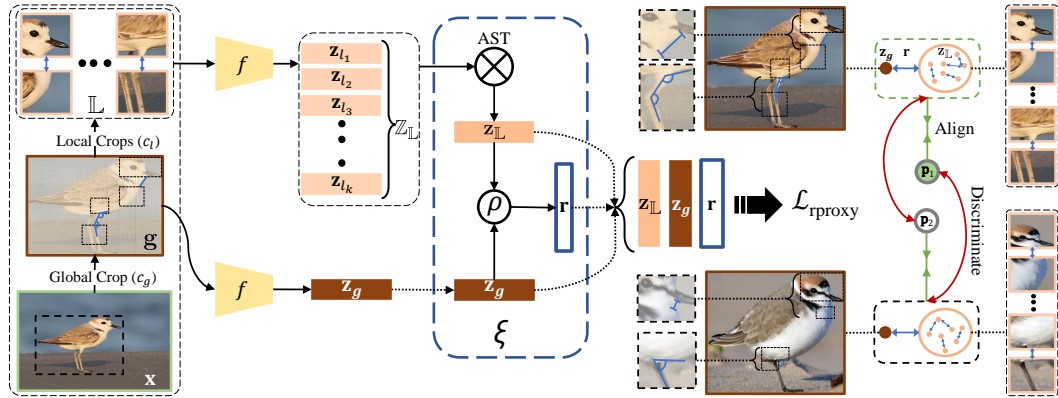

Figure 1: We start by encoding the global and local views using a relation-agnostic encoder $f$. We then compute the cross-view relational embedding $\mathbf{r}$ between the global $\mathbf{z}_g$ and the summary of local $\mathbf{z}_{\mathbb{L}}$ representations. The AST, in conjunction with $\rho$, form the cross-view relational function $\xi$. Finally, the learning of our Relational Proxies is conditioned by both view-specific ($\mathbf{z}_{\mathbb{L}}$ and $\mathbf{z}_g$) and cross-view relational ($\mathbf{r}$) information. Minimizing $\mathcal{L}_{\text{rproxy}}$ helps to align representations from the same category, while discriminating across different categories in a metric space.

## 3.1 Problem Definition

We leverage the qualitative consistency in the definition of the fine-grained visual categorization (FGVC) problem in the relevant literature [25, 48, 13, 3] to formalize the same in more quantitative terms as follows.

**Definition 1** (**k-distinguishability**). *Two categories $\mathcal{C}_1$ and $\mathcal{C}_2$ are said to be k-distinguishable iff along with the global view, a classifier needs at least k local features to tell them apart, i.e., the true hypothesis can only distinguish between $\mathcal{C}_1$ and $\mathcal{C}_2$ if it has access to the complete set $\{\mathbf{z}_g\} \cup \mathbb{Z}_{\mathbb{L}}$, and it fails to distinguish between $\mathcal{C}_1$ and $\mathcal{C}_2$, if it only has access to $\{\mathbf{z}_g\} \cup \mathbb{Z}_{\mathbb{L}} \backslash \mathbf{z}_l, \forall \mathbf{z}_l \in \mathbb{Z}_{\mathbb{L}}$.*

The notion of *k-distinguishability* formalizes what it means for two categories to only be distinguishable in the fine-grained but not in the coarse-grained setting. Given the concept of k-distinguishability, the definition of FGVC problem directly follows from here:

**Definition 2** (**Fine-Grained Visual Categorization Problem** - $\mathcal{P}_{FGVC}$). *A categorization problem is said to belong to the $\mathcal{P}_{FGVC}$ family, iff there exists at least one pair of categories $\mathcal{C}_1$ and $\mathcal{C}_2$ such that they are k-distinguishable.*

Unless otherwise stated, all datapoints $(\mathbf{x}, \mathbf{y})$ are considered to be sampled from $k$-distinguishable categories of an instance of $\mathcal{P}_{\text{FGVC}}$. In the subsequent sections, we prove that for a learner to completely model the class distribution for an instance of $\mathcal{P}_{\text{FGVC}}$, it must, alongside the view specific representations $\mathbf{z}_g$ and $\mathbb{Z}_{\mathbb{L}}$, also learn a function $\xi$ that models the cross-view relationship between the global and the local views. Thus, a function $\xi$, to model $R$, must satisfy the following properties: (1) *View-Unification*: Maps the set of all views $\{\mathbf{g}, \mathbf{l} \in \mathbb{L}\}$ of an image $\mathbf{x}$ to a single output $\mathbf{r}$; (2) *Permutation Invariance*: Produces the same output irrespective of the order of the local attributes, *i.e.*, $\xi(\mathbf{z}_g, \{\mathbf{z}_{l_1}, \mathbf{z}_{l_2}, ... \mathbf{z}_{l_k}\}) = \xi(\mathbf{z}_g, \{\mathbf{z}_{l_{\pi(1)}}, \mathbf{z}_{l_{\pi(2)}}, ... \mathbf{z}_{l_{\pi(k)}}\})$, for every permutation $\pi$, where $\mathbf{z}_g$ and $\mathbf{z}_{l_i}$ are the representations of the global and the local views respectively, obtained from $f$. We provide more details on the necessity of these properties in Section 1 of the Appendix.

## 3.2 Relation-Agnostic Representations and Information Gap

In this section, we formally study the nature of the representation spaces learned by models that do not consider the cross-view relational information in the context of $\mathcal{P}_{\text{FGVC}}$. We term such representations as being "relation-agnostic" and prove via Proposition 1 that they suffer from an Information Gap, and thus are unable to capture the complete label information encoded in an input image.

**Definition 3** (**Relation-Agnostic Representations - Information Theoretic**). *An encoder is said to produce relation-agnostic representations if it independently encodes the global view $\mathbf{g}$ and local views $\mathbf{l} \in \mathbb{L}$ of $\mathbf{x}$ without considering their relationship information $\mathbf{r}$.*

**Lemma 1.** *Given a relation-agnostic representation $\mathbf{z}$ of $\mathbf{x}$, the conditional mutual information between $\mathbf{x}$ and $\mathbf{y}$ given $\mathbf{z}$ can be reduced to $I(\mathbf{x}; \mathbf{r}|\mathbf{z})$.*

*Proof.* Given a relation-agnostic representation $\mathbf{z}$ of $\mathbf{x}$, the only uncertainty that remains about the label information $\mathbf{y}$ can be quantified as the cross-view relational information $\mathbf{r}$, *i.e.*, $I(\mathbf{x}; \mathbf{y}|\mathbf{z}) = I(\mathbf{x}; \mathbf{r})$. The proof of this statement is given in Identity 1 of the Appendix.

Intuitively, the conditional mutual information between $\mathbf{x}$ and $\mathbf{y}$ given $\mathbf{z}$, *i.e.*, $I(\mathbf{x}; \mathbf{y}|\mathbf{z})$ represents the information for predicting $\mathbf{y}$ from $\mathbf{x}$ that $\mathbf{z}$ is unable to capture. Since $\mathbf{z}$ is relation-agnostic, the only uncertainty that remains in $\mathbf{x}$ after $\mathbf{z}$ is the cross-set relationship between the global and the local views, *i.e.*, $\mathbf{r}$. Therefore, we can write $I(\mathbf{x}; \mathbf{y}|\mathbf{z}) = I(\mathbf{x}; \mathbf{r})$. Using this equality and further factorizing $I(\mathbf{x}; \mathbf{r})$ using the chain rule for mutual information, we get:

$$I(\mathbf{x}; \mathbf{y}|\mathbf{z}) = I(\mathbf{x}; \mathbf{r}) = I(\mathbf{x}; \mathbf{r}|\mathbf{z}) + I(\mathbf{r}; \mathbf{z}) = I(\mathbf{x}; \mathbf{r}|\mathbf{z}),$$

the latter equality following from Definition 3, which implies that $I(\mathbf{r}; \mathbf{z}) = 0$, since $\mathbf{z}$ does not explicitly model the local-to-global relationships $\mathbf{r}$. $\qquad\square$

**Lemma 2.** *The mutual information between $\mathbf{x}$ and its relation-agnostic representation $\mathbf{z}$ does not change with the knowledge of $\mathbf{r}$.*

*Proof.* Following the chain rule [12], the mutual information between $\mathbf{x}$ and $\mathbf{z}$, *i.e.*, $I(\mathbf{x}; \mathbf{z})$ can be expressed as $I(\mathbf{x}; \mathbf{z}|\mathbf{r}) + I(\mathbf{z}; \mathbf{r})$. However, since $\mathbf{z}$ is *relation-agnostic* (Definition 3), $I(\mathbf{z}; \mathbf{r}) = 0$. Thus, $I(\mathbf{x}; \mathbf{z}) = I(\mathbf{x}; \mathbf{z}|\mathbf{r}) + I(\mathbf{z}; \mathbf{r}) = I(\mathbf{x}; \mathbf{z}|\mathbf{r})$. $\qquad\square$

**Proposition 1.** *For relation-agnostic representation $\mathbf{z}$ of $\mathbf{x}$, the label information encoded in $\mathbf{z}$ is strictly upper-bounded by the label information in $\mathbf{x}$, i.e., $I(\mathbf{x}; \mathbf{y}) > I(\mathbf{z}; \mathbf{y})$ by an amount $I(\mathbf{x}; \mathbf{r}|\mathbf{z})$.*

*Proof.* The mutual information $I(\mathbf{x}; \mathbf{y})$ between a datapoint $\mathbf{x}$ and its ground-truth label $\mathbf{y}$ can be expressed as $I(\mathbf{x}; \mathbf{y}|\mathbf{z}) + I(\mathbf{x}; \mathbf{z})$ based on the chain rule. Here $I(\mathbf{x}; \mathbf{y}|\mathbf{z})$ represents the information for predicting $\mathbf{y}$ from $\mathbf{x}$ that $\mathbf{z}$ is unable to capture, while $I(\mathbf{x}; \mathbf{z})$ denotes the predictive information that $\mathbf{z}$ does capture from $\mathbf{x}$. We can thus rewrite $I(\mathbf{x}; \mathbf{y})$ using Lemma 1 and Lemma 2 as:

$$I(\mathbf{x}; \mathbf{y}) = I(\mathbf{x}; \mathbf{y}|\mathbf{z}) + I(\mathbf{x}; \mathbf{z}) = I(\mathbf{x}; \mathbf{r}|\mathbf{z}) + I(\mathbf{x}; \mathbf{z}|\mathbf{r}) \tag{1}$$

Now, using the chain rule of mutual information, $I(\mathbf{z}; \mathbf{y}) = I(\mathbf{z}; \mathbf{y}|\mathbf{x}) + I(\mathbf{z}; \mathbf{x})$. However, as a consequence of the data processing inequality [12], $I(\mathbf{z}; \mathbf{y}|\mathbf{x}) = 0$ (since $\mathbf{z}$ cannot encode any more information about $\mathbf{y}$ than $\mathbf{x}$). Applying this and Lemma 2 to Equation (1):

$$I(\mathbf{x}; \mathbf{y}) = I(\mathbf{x}; \mathbf{r}|\mathbf{z}) + I(\mathbf{x}; \mathbf{z}|\mathbf{r}) = I(\mathbf{x}; \mathbf{r}|\mathbf{z}) + I(\mathbf{x}; \mathbf{z}) = I(\mathbf{x}; \mathbf{r}|\mathbf{z}) + I(\mathbf{z}; \mathbf{y})$$

Therefore, $I(\mathbf{x}; \mathbf{y}) > I(\mathbf{z}; \mathbf{y})$, by an amount $I(\mathbf{x}; \mathbf{r}|\mathbf{z})$. $\qquad\square$

**Intuition:** By establishing a strict upper-bound, Proposition 1 shows that relation-agnostic encoders cannot fully capture the label information in an input image. The quantity they are unable to capture is given by $I(\mathbf{x}; \mathbf{r}|\mathbf{z})$, which we call the Information Gap.

### 3.3 Sufficient Learner

Proposition 1 states that the information gap exists *if* the representation space happens to be relation-agnostic. We now explore if there is really the need to learn relation-agnostic representations in the first place. From there, we identify the necessary and sufficient conditions for a complete learning of $I(\mathbf{x}; \mathbf{y})$, and derive the requirements for a learner to do the same.

**Definition 4 (Relation-Agnostic Representations - Geometric).** *Let $n_\epsilon(\cdot)$ represent the $\epsilon$-neighbourhood around a point in the limit $\epsilon \to 0$[2]. A representation space is relation-agnostic if and only if $\forall \mathbf{z}_l \in \mathbb{Z}_L : n_\epsilon(\mathbf{z}_l) \cap n_\epsilon(\mathbf{z}_g) = \phi$.*

An intuitive explanation of Definition 4 can be found in Section 3 of the Appendix.

**Axiom 1.** *$f$ learns representations $\mathbf{z}$ such that a classifier operating on the domain of $\mathbf{z}$ learns a distribution $\hat{\mathbf{y}}$, minimizing its cross-entropy with the true distribution $-\sum_i \mathbf{y}_i \log(\hat{\mathbf{y}}_i)$, where $i$ denotes the $i$-th class.*

**Lemma 3.** *For an instance of $\mathcal{P}_{FGVC}$, the representation space learned by $f$ is relation-agnostic, i.e., the global view $\mathbf{g}$ and the set of local views $\mathbf{l} \in \mathbb{L}$ are mapped to disjoint locations in the representation space.*

---

[2]The choice of $\epsilon$ determines the degree of relation-agnosticity of the representation space.

*Proof.* From Definition 4, a representation space is not relation-agnostic *iff* $\exists \mathbf{z}_l \in \mathbb{Z}_L : n_\epsilon(\mathbf{z}_l) \cap n_\epsilon(\mathbf{z}_g) \neq \phi$. Under this condition, the classifier only has the information from $\{\mathbf{z}_g\} \cup \mathbb{Z}_L \backslash \mathbf{z}_l$ instead of the required $\{\mathbf{z}_g\} \cup \mathbb{Z}_L$. Thus, for instances of $\mathcal{P}_{\text{FGVC}}$, according to Definition 1, removing the relation-agnostic nature from the representation space of $f$ would cause a downstream classifier to produce misclassifications across the instances of $k$-distinguishable categories, leading to a violation of Axiom 1. Hence, $f$ can only learn relation-agnostic representations. $\square$

We can thus conclude from Lemma 3 and Proposition 1 that the necessary and sufficient conditions for a learner to capture the complete label information $I(\mathbf{x}; \mathbf{y})$, are to consider both (1) the relation-agnostic information $\mathbf{z}$ and (2) the cross-view relational information $\mathbf{r}$.

**Proposition 2.** *An encoder $f$ trained to learn relation-agnostic representations $\mathbf{z}$ of datapoints $\mathbf{x}$ cannot be used to model the relationship $\mathbf{r}$ between the global and local views of $\mathbf{x}$.*

*Proof.* $f : \mathbf{x}_v \to \mathbf{z}_v$ is a unary function that takes as input a (global or local) view $\mathbf{x}_v$ of an image $\mathbf{x}$ and produces view-specific (Lemma 3) representations $\mathbf{z}_v$ for a downstream function $g : \mathbf{z}_v \to \mathbf{y}$.

For $f$ to model the cross-view relationships, it must output the same vector $\mathbf{r}$ irrespective of whether $\mathbf{x}_v = \mathbf{g}$ or $\mathbf{x}_v = \mathbf{l} \in \mathbb{L}$, *i.e.* whether $\mathbf{x}_v$ is a global or a local view of the input image $\mathbf{x}$ (*view-unification* property of $\xi$). However, Lemma 3 prevents this from happening by requiring the output space of $f$ to be relation-agnostic. Hence, $f$ cannot be used to model $\mathbf{r}$. $\square$

Thus, to bridge the information gap, a learner must have distinct sub-models that individually satisfy the properties of being relation-agnostic and relation-aware. Only such a learner could qualify as being sufficient for an instance of $\mathcal{P}_{\text{FGVC}}$.

**Intuition:** In this section, we have effectively proven that the properties of relation-agnosticity and relation-awareness are dual to each other. We show that while relation-agnosticity is not sufficient, it is a necessary condition for encoding the complete label information $I(\mathbf{x}; \mathbf{y})$. We also show that a disjoint encoder cannot be used to model the two properties alone without violating one of the necessary criteria. The requirement of a separate, relation-aware sub-model follows from here.

### 3.4 Learning Relation-Agnostic and Relation-Aware Representations

Figure 1 depicts the end-to-end design of our framework. Derived from our theoretical findings, it comprises of both the relation-agnostic agnostic encoder $f$, and the cross-view relational function, $\xi$, expressed as a composition of the Attribute Summarization Transformer, AST, and a network for view-unification, $\rho$. Below, we elaborate on each of these components.

**Relation-Agnostic Representations:** We follow recent literature [43, 48] for localizing the object of interest in the input image $\mathbf{x}$ and obtaining the global view $\mathbf{g}$ by thresholding the final layer activations of a CNN encoder $f$ and detecting the largest connected component in the thresholded feature map. We obtain the set of local views $\{\mathbf{l}_1, \mathbf{l}_2 \ldots, \mathbf{l}_k\}$ as sub-crops of $\mathbf{g}$ (more details in Section 4.1). Following the primary requirement of Proposition 1, we produce relation-agnostic representations by propagating $\mathbf{g}$ and $\mathbf{l}_i$ through a CNN encoder $f$ that independently encodes the two view families as $\mathbf{z}_g = f(\mathbf{g})$ and $\mathbf{z}_{l_i} = f(\mathbf{l}_i)$.

**Relational Embeddings:** The second requirement, according to Proposition 1, for completely learning $I(\mathbf{x}; \mathbf{y})$ is to minimize $I(\mathbf{x}; \mathbf{r}|\mathbf{z})$, *i.e.*, the uncertainty about the relational information $\mathbf{r}$ encoded in $\mathbf{x}$, given a relation-agnostic representation $\mathbf{z}$. However, according to Proposition 2, we cannot perform the same using the relation-agnostic encoder $f$. Contrary to existing relational learning literature [31, 33] that assumes the nature of relationships to be known beforehand, we take a novel approach that models cross-view relationships as learnable representations of the input $\mathbf{x}$. We follow the definition of the relationship modelling function $\xi : (\mathbf{g}, \mathbb{L}) \to \mathbf{r}$, that takes as input relation-agnostic representations of the global view $\mathbf{z}_g$ and the set of local views $\mathbb{Z}_\mathbb{L} = \{\mathbf{z}_{l_1}, \mathbf{z}_{l_1}, \ldots \mathbf{z}_{l_k}\}$, and outputs a relationship vector $\mathbf{r}$, satisfying the *View-Unification* and *Permutation Invariance* properties.

We satisfy the Permutation Invariance property by aggregating the local representations via a novel Attribute Summarization Transformer (AST). We form a matrix whose columns constitute a learnable summary embedding $\mathbf{z}_\mathbb{L}$ followed by the local representations $\mathbf{z}_{l_i}$ as $\mathbf{Z}'_\mathbb{L} = [\mathbf{z}_\mathbb{L}, \mathbf{z}_{l_1}, \mathbf{z}_{l_1}, \ldots \mathbf{z}_{l_k}]$. We compute the self-attention output $\mathbf{z}'_*$ for each column $\mathbf{z}_*$ in $\mathbf{Z}'_\mathbb{L}$ as $\mathbf{z}'_* = \mathbf{a} \cdot \mathbf{Z}_\mathbb{L} \mathbf{W}$, where $\mathbf{a} = \sigma\left((\mathbf{z}_* \mathbf{W}_q) \cdot (\mathbf{Z}_\mathbb{L} \mathbf{W})^T / \sqrt{D}\right)$, and $D$ is the embedding dimension. By iteratively performing

self-attention operations among the columns of $\mathbf{Z}'_{\mathbb{L}}$, AST aggregates information across all the local attributes into the final learnable output of $\mathbf{z}_{\mathbb{L}}$. Unlike the usual vision transformer [40], we omit the usage of positional embeddings, as doing so provides better permutation invariance [29].

For satisfying the View-Unification property, we introduce a simple feed-forward multilayer perceptron that learns the mapping $\rho : (\mathbf{z_g}, \mathbf{z_{\mathbb{L}}}) \rightarrow \mathbf{r}$. It takes as input the representation of the global view $\mathbf{z}_g$ and the summary of the set of local views $\mathbf{z}_{\mathbb{L}}$, and outputs the relationship as a learned vector $\mathbf{r}$. Thus, in our construction, the AST along with $\rho$, constitute the relation modelling function $\xi$.

**Learning Relational Proxies:** The representations $\mathbf{z}_g, \mathbf{z}_{\mathbb{L}}$ and $r$ in unison encode the full semantic information $\mathbf{y}$ in $\mathbf{x}$ (Proposition 1). To alleviate the low inter-class variance in $\mathcal{P}_{\text{FGVC}}$, metric learning has been shown to be an effective [7] approach. Furthermore, approaches like [28] and [18] for metric learning have shown that substituting pairwise comparisons with assignment to a fixed set of learnable class proxies reduces the training-time complexity from a large polynomial like $\mathcal{O}(n^2)$ or $\mathcal{O}(n^3)$ to near linear $\mathcal{O}(c.n)$, where $c$ is the number of classes in a dataset, $n$ is the number of train-set datapoints, and $c \ll n$. For this purpose, we contrast instance representations across classes through class proxy vectors that are informed by both the view specific and relational representations via learning the conditional distribution $p(\mathbf{y}|\mathbf{z}_g, \mathbf{z}_{\mathbb{L}}, \mathbf{r})$. We term such class proxies, Relational Proxies, as they leverage cross-view relationship information for encoding class semantics.

Consider a set of $c$ learnable class proxy vectors $\mathbb{P} = \{\mathbf{p}_1, \mathbf{p}_2, ... \mathbf{p}_c\}$, where $c$ is the number of fine-grained classes. Here, we present a novel formulation of the proxy-anchor loss [18] in cross-entropic terms that allows us to conform to the requirement of Axiom 1 in the fine-grained setting. Specifically, for each of the representations $\omega \in \{\mathbf{z}_g, \mathbf{z}_l, \mathbf{r}\}$ for all $\mathbf{x} \in \mathbb{X}$, we minimize the following:

$$\mathcal{L}_{\text{rproxy}} = -\frac{1}{c} \sum_{\mathbf{p} \in \mathbb{P}} \log \left( \frac{1}{\psi^+ \cdot \psi^-} \right), \quad \begin{cases} \psi^+ = 1 + \sum\limits_{\omega \in \Omega} e^{-\alpha(s(\omega, \mathbf{p}) - \delta)} \\ \psi^- = 1 + \sum\limits_{\omega \in \bar{\Omega}} e^{\alpha(s(\omega, \mathbf{p}) + \delta)} \end{cases} \tag{2}$$

where $\Omega$ is the set of representations in a mini-batch for which $\mathbf{p}$ is the true class proxy, $\bar{\Omega}$ is one for which $\mathbf{p}$ is not the true class proxy, and $s(\cdot, \cdot)$ computes the cosine distance. $\psi^+$ helps align matching $(\omega, \mathbf{p})$ pairs close together in the representation space (since $\mathcal{L}_{\text{rproxy}}$ follows a cross-entropic form, it does not violate the relation-agnosticity of $f$, as proven in Lemma 3, with a more detailed note in Section 4 of the appendix), while $\psi^-$ helps embedding non-matching $(\omega, \mathbf{p})$ pairs farther apart. The scaling parameter $\alpha$ along with the margin parameter $\delta$ control the intensity with which the alignment and discrimination are performed. $1/(\psi^+ \cdot \psi^-)$ gives a probability indicating how closely the learned representation space reflects the semantic structure in $\mathbb{X}$. $\mathcal{L}_{\text{rproxy}}$ thus computes the cross-entropy loss between the ground-truth and the predicted class distributions over the set of proxies.

**Inference:** Given an input image $\mathbf{x}$, we compute its global ($\mathbf{z}_g$), summary of local ($\mathbf{z}_{\mathbb{L}}$), and relational ($\mathbf{r}$) representations using $f$, AST and $\rho$ as explained above. We then predict the class probability distribution $\hat{\mathbf{y}}$ of these representations by computing their soft-assignment scores across the relational proxies. The assignment score for each proxy $\mathbf{p} \in \mathbb{P}$ is computed as follows:

$$\hat{\mathbf{y}}_{\mathbf{p}} = \sum_{\omega \in \{\mathbf{z}_L, \mathbf{z}_g, \mathbf{r}\}} \frac{e^{s(\omega, \mathbf{p})}}{\sum\limits_{\mathbf{p}' \in \mathbb{P}} e^{s(\omega, \mathbf{p}')}}$$

The class corresponding to the relational proxy with the highest assigned score is returned as the prediction.

## 4 Experiments

We now present the implementation details of Relational Proxy, and the results obtained upon evaluating it on benchmark FGVC datasets. We also discuss observations from ablation studies that we performed to validate our theoretical foundations, as well as the implementation specific choices that we made, along with qualitative visualizations of the learned cross-view local relationships.

### 4.1 Experimental Settings and Datasets

**Implementation details** – We implement our Relational Proxy model using the PyTorch [32] deep learning framework, on an Ubuntu 20.04 workstation with a single NVIDIA GeForce RTX 3090 GPU,

| Method | Benchmark | | | | | Cultivar | |
|---|---|---|---|---|---|---|---|
| | FGVC Aircraft | Stanford Cars | CUB | NA Birds | iNaturalist | Cotton | Soy |
| MaxEnt [11] NeurIPS'18 | 89.76 | 93.85 | 86.54 | - | - | - | - |
| DBTNet [51] NeurIPS'19 | 91.60 | 94.50 | 88.10 | - | - | - | - |
| StochNorm [19] NeurIPS'20 | 81.79 | 87.57 | 79.71 | 74.94 | 60.75 | 45.41 | 38.50 |
| MMAL [48] MMM'21 | 94.70 | 95.00 | 89.60 | 87.10 | 69.85 | 65.00 | 47.00 |
| FFVT [42] BMVC'21 | 79.80 | 91.25 | 91.65 | 89.42 | 70.30 | 57.92 | 44.17 |
| CAP [3] AAAI'21 | 94.90 | 95.70 | 91.80 | 91.00 | - | - | - |
| TransFG [13] AAAI'22 | 80.59 | 94.80 | 91.70 | 90.80 | 71.70 | 45.84 | 38.67 |
| **Ours (Relational Proxy)** | **95.25** ± 0.02 | **96.30** ± 0.04 | **92.00** ± 0.01 | **91.20** ± 0.02 | **72.15** ± 0.03 | **69.81** ± 0.04 | **51.20** ± 0.02 |

Table 1: Comparison of classification accuracies obtained by our method (averaged over 5 independent runs) on standard FGVC datasets with current state-of-the-art approaches.

an 8-core Intel Xeon processor and 32 GBs of RAM. Since we proposed a proxy-based approach for learning the relational metric space, we do not have a dependency on batch-size for the purpose of negative sampling as part of our metric learning phase, which enables us to train our entire model end-to-end on a single GPU. Also by the virtue of using class-proxies, the convergence time is reduced by a significant amount compared to pairwise losses.

**Hyperparameter settings** – For initial training stability, we consider five disjoint locations (four corners and the centre) of $g$ to be the set of local views. As training progresses, we also allow the model to learn from an increased number views obtained via random cropping. In the same way at inference time, the local views constitute a combination of the five disjoint crops along with some random crops. We found that the optimal number of local views $k$ to be equal to 7 for FGVC Aircraft, Stanford Cars and both the cultivar datasets. For CUB and NABirds, $k = 8$ gave the best performance. We use ResNet50 [14] pretrained on ImageNet [9] as the backbone of our relation-agnostic encoder $f$. In Sec. 1.3 of the supplementary, we also provide evaluations using VGG-16 [36] to show that the performance gains achieved by our model do not depend on the specific backbone. We train our full Relational Proxy model end-to-end for 200 epochs using the stochastic gradient descent optimizer with an initial learning rate of 0.001 (decayed by a factor of 0.1 every 50 epochs), a momentum of 0.9, and a weight decay of $10^{-4}$.

**Datasets and Evaluation** – We evaluate our model on the four most common fine-grained visual categorization benchmarks (number of classes and train/test splits respectively in brackets): FGVC Aircraft [26] (100 | 6667/3333), Stanford Cars [20] (196 | 8144/8041), CUB [41] (200 | 5994/5794), and NA Birds [39] (555 | 23,929/24,633). For large scale benchmark evaluation, we choose the iNaturalist 2017 dataset which consists of 13 super-categories that have been split into a total of 5089 fine-grained categories with 675,170 training and 182,707 test images. We also perform experiments on two challenging datasets of the cultivar domain that offer very low inter-class variations, namely Cotton Cultivar [46] (80 | 240/240) and Soy Cultivar [46] (200 | 600/600). We use classification accuracy as our metric for evaluating the performance of a model.

### 4.2 Comparison with State of the Art

**Benchmark Datasets** – In Table 1, we report the performance of our method on benchmark datasets along with existing SotA approaches. StochNorm [19] presents a novel way to refactor batch normalization that helps prevent overfitting for the task of FGVC. The novel training routine proposed in MaxEnt[11] improves FGVC performance by maximizing the entropy of the output probability distribution of a CNN. By designing a computationally inexpensive bilinear feature transformation mechanism for CNNs, DBT [51] achieves competitive performance on benchmark FGVC datasets. MMAL [48] is one of the most competitive models for FGVC Aircraft and Stanford Cars, which extracts the most informative global and local views by analyzing the activation maps of the final layer of a CNN, and embeds them in a relation-agnostic representation space. TransFG [13] proposes a vision transformer based technique for extracting informative local patches, achieving SotA performance on iNaturalist, and promising results on CUB and NA Birds. By learning a context aware attention pooling mechanism, CAP [3] reports SotA performance on all benchmark datasets other than iNaturalist. From Table 1, we see that our method surpasses the SotA on all four benchmarks by significant margins. Specifically, we beat the SotA on Stanford Cars by $0.60\%$, on iNaturalist by $0.45\%$, on FGVC Aircraft by $0.35\%$, and on both CUB and NA Birds by $0.20\%$.

Cars and Aircrafts can largely vary in color, texture and custom, part-specific styles within a category. However, the geometry of the overall object (represented by cross-view relationships) within a class remains fairly constant. This leaves room for a large amount of relational information to be captured. This also holds true for the iNaturalist dataset, as the local-to-global emergent relationships can be used to discriminate between both coarse-grained (super) and fine-grained (sub) categories. For the

| ID | Relation-Agnostic Encoder | AST | RelationNet | Learnable Relation | Proxies | Aircraft | CUB | Stanford Cars |
|---|---|---|---|---|---|---|---|---|
| 1. | ✓ | | | | | 94.60 | 91.25 | 95.21 |
| 2. | ✓ | ✓ | | | | 94.91 | 91.50 | 95.62 |
| 3. | ✓ | ✓ | ✓ | ✓ | | 95.13 | 91.90 | 96.15 |
| 4. | ✓ | | | ✓ | ✓ | 94.92 | 91.55 | 95.70 |
| 5. | ✓ | ✓ | | ✓ | ✓ | 95.10 | 91.81 | 96.05 |
| 6. | ✓ | | ✓ | ✓ | ✓ | 95.05 | 91.73 | 95.93 |
| 7. | ✓ | ✓ | ✓ | ✓ | ✓ | **95.25** | **92.00** | **96.30** |

Table 2: Results of ablating the key components of our Relational Proxy model (sufficient learner). Grouped so as to better illustrate the effect of learning cross-view relationships. Note that the meaning of non-existence of a component may vary according to context / other row elements. Refer to the corresponding paragraph in Section 4.3 of the main text for further details.

| ID | Attribute Summary ($\mathbf{z}_{\mathbb{L}}$) | Global Representation ($\mathbf{z}_g$) | Relational Representation ($\mathbf{r}$) | FGVC Aircraft | CUB | Stanford Cars |
|---|---|---|---|---|---|---|
| 1. | ✓ | ✓ | | 94.91 | 91.50 | 95.62 |
| 2. | ✓ | | ✓ | 94.85 | 91.58 | 95.75 |
| 3. | | ✓ | ✓ | 94.60 | 91.47 | 95.51 |
| 4. | ✓ | ✓ | ✓ | **95.25** | **92.00** | **96.30** |

Table 3: Results of ablating the Proxy Conditioning Representations. Note that only the output representation vectors were ablated here, and not the entire model component producing them. The latter has been studied independently with findings reported in Table 2.

bird datasets (CUB, NABirds), although this relational information is still there, most categories can be told apart by color, texture and local-attribute specific information, if they are clearly visible. For this reason, the accuracy gains obtained in the Cars and Aircraft datasets surpass those obtained for the birds.

**Cultivar Datasets** – For the highly challenging datasets of the cultivar domain, *i.e.*, Cotton and Soy Cultivar, FFVT [42] provides state-of-the-art results by using a specialized feature fusion technique for vision transformers. As can be seen in Table 1, our model, by leveraging cross-view relational embeddings, manages to provide a performance boost exceeding $4\%$ over the current SOTA on the cultivar datasets. Cultivar datasets have very low inter-class differences. Cross-view relational information like edge curvature, relative angles between leaf sub-parts, width to height ratio, convergence patterns of leaf ends, etc., largely determine the uniqueness of a category. For this reason, our method is extremely effective when applied to such domains.

### 4.3 Ablation Studies

We perform the following three classes of ablation studies:

**Key components of the sufficient learner** – Table 2 shows the results of ablating the key components of our model. The relation agnostic encoder being the most fundamental component, cannot be removed, and therefore appears in all the rows. Row 1 thus represents training a simple classification head on top of the representations obtained from the relation-agnostic encoder. Row 2 denotes the result of aggregating the local views, computing a predefined relationship function, specifically the distance between the local and global representations, and minimizing a Huber loss between the relational distance value between instances of the same class. Row 3 introduces the idea of learnable relational vectors (instead of predefined functions like distances). Since cross-view relationships are unique to a class, we aim to embed the relational vectors in a metric space by minimizing a pairwise contrastive loss across classes. However, as noted in recent metric learning literature [28, 18], computing pairwise losses can be expensive and lead to slower convergence. Motivated by this, we introduce the idea of relational proxies in Rows 4 - 7. Row 4 replaces the AST and RelationNet by simple concatenation of the inputs and propagation through a linear layer. Row 5 and 6 individually show the effects of replacing the AST and RelationNet with linear layers. Finally, Row 7 denotes the performance of our model with all components included.

Rows 2 and 3 demonstrate the importance of modelling cross-view relationships specifically as a learnable metric space embedding. Rows 5, 6 and 7 show the contribution of our AST in summarizing the local attributes, as well as the fact that the cross-view relationship is non-linear in nature.

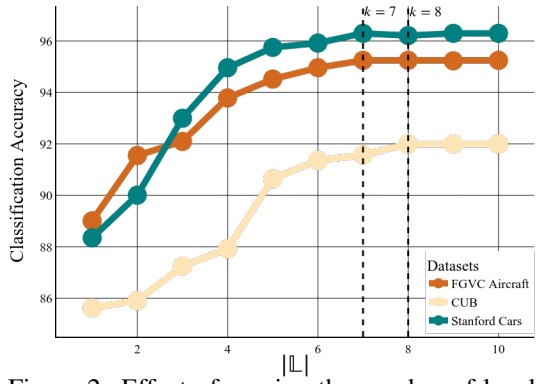

| Method | T-ImageNet | D-ImageNet |
|---|---|---|
| $\mathbf{z}_g, \mathbf{z}_{\mathbb{L}}$ | 88.75 | 91.30 |
| $\mathbf{z}_g, \mathbf{z}_{\mathbb{L}}, \mathbf{r}$ | 88.91 | 92.75 |
| $\Delta$ | 0.16 | **1.45** |

Table 4: Comparison of accuracy gains obtained by using the relational information $\mathbf{r}$ along with the view-specific representations $\mathbf{z}_g$ and $\mathbf{z}_{\mathbb{L}}$ (second row) over using only relation-agnostic representations (first row) on the coarse-grained Tiny (T) and fine-grained Dogs (D) subsets of ImageNet. As can be observed, the performance gain $\Delta$ by accounting for the relational information is significantly more in the fine-grained setting than in the coarse-grained one.

Figure 2: Effect of varying the number of local views $|\mathbb{L}|$

**Conditioning of the relational proxies** – The relational proxies in our model are conditioned by three representations of the input $\mathbf{x}$ – the summary of the local attributes $\mathbf{z}_{\mathbb{L}}$, the representation of the global view $\mathbf{z}_g$, and the relational vector $\mathbf{r}$. We study the contribution of each of these representations and summarize our findings in Table 3. These results demonstrate that the information encoded in all three representations are necessary for learning the complete set of class attributes.

**Results on ImageNet subsets** – In order to validate whether our findings are in fact particularly applicable to the fine-grained setting, we perform experiments to compare the performance boost provided by our method over a vanilla relation-agnostic encoder, between coarse-grained (Tiny ImageNet[22]) and fine-grained (Dogs ImageNet / Stanford Dogs [17]) subsets of ImageNet. Our findings are summarized in Table 4, which shows that our method does in fact provide a more significant improvement over a relation-agnostic encoder in the fine-grained setting.

**The optimal value of $k$ for the $k$-distinguishability criterion** – Figure 2 empirically illustrates the idea of $k$-distinguishability for a given local-crop size on the FGVC Aircraft, Stanford Cars and CUB datasets. For an instance of $\mathcal{P}_{\text{FGVC}}$, the performance of a model is strongly dependent on the number of local views it has access to. When the number of local views $|\mathbb{L}|$ is less than the minimum required number $k$, the classification performance is poor as the model does not have access to the minimum set of required fine-grained information. As $|\mathbb{L}|$ approaches $k$, the performance increases, reaching its maximum at $k$ (= 7 for FGVC Aircraft and Stanford Cars, and 8 for CUB). However, if $|\mathbb{L}|$ is increased beyond $k$, there is no further gain in performance, as the extra information is either redundant or semantically irrelevant.

**Correlation between $|\mathbb{L}|$ and local patch size** – To determine the right computational trade-offs for our method, we perform a study to identify possible correlations between the number of local views $\mathbb{L}$ and size of local patches. We trained our model on FGVC Aircraft [26] by varying the number of local views $|\mathbb{L}|$ and the size of each local patch to identify their correlations. We present our results in Table 5, where rows represent the number of local views $|\mathbb{L}|$ and the columns represent the side-length of each local patch. So, if the global view has spatial dimensions $N \times N$, each local patch would be of $N/t \times N/t$, where $t$ is the scaling factor that is varied across the columns. In summary, the rows represent increasing the number of local views top-down, and the columns represent increasing the patch-size left-to-right. The numbers are expressed as relative deviations from a reference of 95.25%, i.e., the setting corresponding to our reported accuracy for FGVC Aircraft in Table 1.

From Table 5, we can see that increasing the patch size beyond a certain point has a detrimental effect as the local views tend to lose their granularity and degenerate into global views. Increasing the number of crops has a stronger improvement effect on performance if the patch size is small, thus influencing the value of $k$ accordingly. However, decreasing the patch size at the cost of an increased number of local views also has its downsides - the number of attention computations in the attribute summarization step increases quadratically. Thus $|\mathbb{L}|$ and the local patch size needs to be determined based

| $\mathbb{L}$ | N/5 | N/4 | N/3 | N/2 |
|---|---|---|---|---|
| **7** | -0.03 | -0.02 | 0.00 | -0.14 |
| **12** | +0.01 | +0.02 | 0.00 | -0.11 |
| **15** | +0.05 | +0.03 | +0.01 | -0.11 |
| **18** | +0.05 | +0.03 | +0.00 | -0.10 |

Table 5: Correlation between number of local views and size of local patches

on application specific accuracy requirements and the available computational resources.

### 4.4 Visual Representations of Cross-View Local Relationships

Our AST-based aggregation scheme allows us to visualize the local relationships that lead to the emergence of the global-view. We aim to construct a graph of local views for depicting the cross-view local relationships. The graph represents the manner in which the local views combine to form the overall object. The nodes of the graph represent the local views. Two nodes are connected via an edge if there exists a relationship between them. The thickness of the edges in the illustration is proportional to the degree of relatedness. We compute the topology of this graph by analyzing the final layer mutual attention values of the Attribute Summarization Transformer (AST). We add an edge between two local views if their mutual attention score is higher than a threshold (which we choose to be the average of all pairwise attention scores). The weight of the edge is proportional to the magnitude of attention. For the purpose of simplicity, we depict fewer local views in the visualization, than are actually used for computation. Figure 3

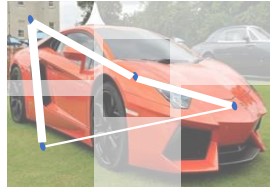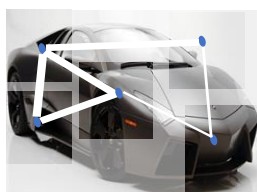

Lamborghini Aventador      Lamborghini Reventón

Figure 3: Visualization of the learned relationships across local-views. Despite close similarities between the two car categories, it can be seen that our model leverages discriminative cross-view relationships to tell the instances apart.

shows example graphs on images from the Stanford Cars dataset. In Appendix Section 5, we provide more such qualitative results and based on these graphs, we provide an analysis of scenarios under which even relational information cannot distinguish between certain fine-grained categories.

## 5 Conclusion and Discussion

Starting with the idea of $k$-distinguishability, we derived the necessary and sufficient conditions that a model must satisfy in order to completely capture the fine-grained information in an image. We proved that a model needs to simultaneously encode both view-specific and cross-view relational properties of an object in order to bridge the information gap that its representations have with the semantic content in the input image. Based on our theoretical findings, we designed Relational Proxies, a method that achieves state-of-the-art results on benchmark FGVC datasets by learning class representations conditioned with cross-view relationships. By introducing a theoretically rigorous framework, we believe that our work opens up new avenues for studying the problem of FGVC in a more systematic manner. One immediate potential outcome of our work that we foresee is the development of explainable fine-grained features. Such features can be used for computing a minimal set of fine-grained attributes to limit compute time/resources, or to perform tasks like cross-modal retrieval in domains with large modality gap.

**Limitations** – The process of obtaining local views in our method is somewhat of an uninformed, generic cropping methodology on the global view of the object, which may not necessarily always yield the best set of local object parts. More informed ways of detecting novel object parts from which the global view emerges can lead to obtaining at par performance but with fewer local views.

**Societal Impacts** – The rigorous theoretical basis of our work has a positive societal impact, which not only makes our methodology transparent and easy to analyze, but also provides a framework to study the foundations of FGVC in general. So far, we are not aware of any negative societal impact that is specific to our methodology. However, as with all data-driven approaches, underlying biases in the datasets on which our model is trained would influence the patterns learned by it.

## Acknowledgements

This work has been partially supported by the ERC 853489–DEXIM, by the DFG–EXC number 2064/1–Project number 390727645, and as part of the Excellence Strategy of the German Federal and State Governments.

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
