# OpenReview forum: "Relational Proxies: Emergent Relationships as Fine-Grained Discriminators"
_NeurIPS.cc/2022/Conference — NeurIPS 2022 Accept_

### Official Review · Reviewer_erdD · 2022-07-06

**Rating:** 6
**Confidence:** 3
**Soundness:** 2 fair
**Presentation:** 2 fair
**Contribution:** 2 fair

**Summary:**

This paper proposes a method based on relationships between views of an object to perform fine grained visual categorization. The authors hypothesise that not only representing local parts but relating them is pivotal to achieve good performance. The authors then experiment using their proposed approach on different FGVC datasets, improving on current methods.

**Questions:**

Stated above.

**Limitations:**

The authors discuss limitations.

**Strengths And Weaknesses:**

Strengths:
1. The point the paper makes, that the relationship between object parts is important for learning the classification label makes intuitive sense and seems to be important in their experiments.

2. The paper is reasonably clear as to the motivations and experiments run by the authors.

Weaknesses
1. The authors, in my opinion, over complicate the mathematical explanation, spending two pages explaining why modelling the relationships between object parts is important. I do not think this adds much to the paper and the space would be better spent giving a high level, clear explanation of the intuition.

2. I am not sure why their AST (how they combine local parts) is so different from an attention layer (e.g. in TransFG). It seems to be performing the same operation, so I do not understand how this is fundamentally different than the TransFG architecture. Is the main difference the global embedding of the object that the local crops can be compared against and using all three embeddings -- the z_g, z_L, r -- when computing distances and the final metric?

3. In general the paper is clear: their main contribution is using the global, local and r information in the metric learning and so the insight is that using all 3 sources of information is the most useful. However, in the experiments there seems to be limited improvement from these properties over the base encoder (Table 2). So I wonder how useful these things really are. Moreover, the improvements are small over standard methods, and there is no standard deviation to explain if these results are significant. I further wonder if the authors carefully made sure that their setup was similar to the underlying setup and that the improvement was actually due to their method or better data augmentation or underlying architectures.

---

> ### Author Response · Authors · 2022-08-02
> **Response to Reviewer erdD**
>
> 1. **Complicated mathematical modelling and explanation:** We thank the reviewer for the suggestion. However, one of the core motivations behind our paper is to provide a theoretical framework for the analysis of FGVC algorithms, something that we felt was missing in existing literature. We have tried to accomplish this by making the mathematical foundations of our paper as comprehensive as we could (also observed by Reviewers 1ZBu, 9xUV, EFjT, and G6W7). At the same time, we acknowledge that this recommendation is helpful, and we will incorporate in the main text, intuitive explanations (some of which are currently in the appendix - Sections 6.2 and 6.3) for the underlying theory.
>
> 2. **AST:** As correctly observed by the reviewer, the AST being a Transformer, has the attention operation as its building block. However, our task requires it to be permutation invariant, for which it can no longer depend on the position embeddings, which happens to be an important and unique design decision specific to our particular problem formulation. However, **please note that we claim novelty not so much for its design, but rather the objective with which it is trained**. Conventional ViTs for FGVC (like TransFG) are trained so as to base their classification only on the information provided by the most discriminative image patches. In our case, the purpose that AST serves is to exhaustively enumerate and summarize all the possible ways the set of local views can combine to form the global view. In other words, it learns to enumerate all possible local-to-global relationships that the set of local views can generate. This is achieved by the virtue of its permutation invariance.
> The view-unification MLP ($\rho$) then matches the correct relationship from this enumerated set to the global-view embedding. We ensure that the enumerations produced by the AST are semantically meaningful by matching them to the correct relational proxy of the input image. We have elaborated further on the working principle of our AST in our answer to Reviewer EFjT’s Question 1 on its necessity of being permutation invariant, which we will include in the final version of the paper.
>
> 3. **Standardized experimental setup:** We follow the same experimental setup as the standard FGVC literature that we compare our work to [3,11,47,50], which involve ResNet50 as the network backbone and random horizontal flip, along with color jitter as the set of data augmentations. This gives us a fair starting point to quantify the gains achieved specifically by our method.
> Also, as was suggested by Reviewer 9xUV, we have now evaluated our model with the VGG-16 backbone on the FGVC Aircraft and CUB datasets. We present our results below with error bars, and compare them to existing SotA methods that also provide their accuracies on VGG-16.
>
> ||FGVC Aircraft|CUB
> |-|:-:|:-:|
> MaxEnt [5]|78.08|77.02
> MMAL [6]|87.00|83.75
> Ours|**91.20** ± 0.03|**88.13** ± 0.01
>
> As the numbers show, our method has no backbone specific dependency, providing stable and consistent improvements over SotA across different underlying architectures.\
> **Standard deviations / Error bars:** We thank the reviewer for this suggestion. We will add the error bars (over 5 independent runs with different random seed initializations) for Table 1 in the final version of the paper. For completeness, we also present them below:
>
> |Dataset | Accuracy (mean ± std) |
> |-: | :-: |
> |FGVC Aircraft| 95.25 ± 0.02 |
> |Stanford Cars| 96.30 ± 0.04 |
> |CUB| 92.00 ± 0.01 |
> |NA Birds| 91.20 ± 0.02 |
> |Cotton| 69.81 ± 0.04 |
> |Soy| 51.20 ± 0.02 |
>
> It can be seen that our method does in fact give consistent and stable performance gains across multiple initializations, with a highest error bound of only ± 0.04% across all 6 datasets.\
> **Improvement over ablation baseline:** In our Table 2 of ablation studies, the base encoder / relation-agnostic encoder (row 1) is trained with all the state-of-the-art design choices in terms of network backbone, data-augmentations, hyperparameters, representation learning mechanism, etc. For that reason, **it can be seen that the relation-agnostic encoder itself performs at par with the SotA methods, often exceeding some of the slightly older approaches**. With this very motivation, we designed the theoretical framework of Relation Agnosticity, along with its corresponding experimental counterpart, which serves both as a unified representative for state-of-the-art, as well as a baseline for our method. Thus, our Relation Agnostic Encoder can also serve as a self-contained FGVC method in and of itself. However, we aim to show that even with an encoder that operates at its full capacity when processing views in isolation, it is possible to provide an extra boost to its performance by modelling the cross-view relationships. The gains are more significant, i.e., exceeding 4%, for the datasets on which the SotA encoders do not perform as well.

---

> > ### Comment · Reviewer_erdD · 2022-08-08
> > **Response to rebuttal**
> >
> > I appreciate the efforts the authors went through in the rebuttal. They added new experiments, error bars, clarifications, etc.
> >
> > In light of the new experiments and the explanations offered to me and other reviewers I have more confidence in the correctness and motivations for their work. It seems that the relational representation is useful for performing fine grained classification (though the base encoders seem most useful). However, the additional boost may be useful in some applications and allow for classification in new settings. I hope the authors include a bit more intuition of their theoretical framework in the writing and I have updated my score.

---

> > > ### Author Response · Authors · 2022-08-08
> > > **Note of thanks**
> > >
> > > We thank the reviewer for recognizing our efforts and increasing the score. As per their suggestion, we will incorporate intuitive explanations for our theoretical framework (some of which are currently in the appendix - Sections 6.2 and 6.3) in the final version of our main paper.

---

### Official Review · Reviewer_G6W7 · 2022-07-11

**Rating:** 7
**Confidence:** 3
**Soundness:** 3 good
**Presentation:** 3 good
**Contribution:** 3 good

**Summary:**

In this paper, authors propose to address the fine-grained image classification from a novel perspective, namely for the fine-grained classes, with the visually similar local features, more attention should be made on leveraging the relational information between the global and local views of an object for encoding its semantic label. Relational proxies are designed based on the proposed theory and achieve the superior results on multiple benchmark datasets.

**Questions:**

As noted in the weakness part, I'd like to see some visualized validations of the hypothesis to verify the effectiveness and limitations of the proposed method.

**Limitations:**

As noted by authors, the main limitation is considered as the local views obtained by the proposed method is cropped from the global view, which may not be the best representations of local parts.

**Strengths And Weaknesses:**

Strengths:
A hypothesis is made for fine-grained visual classification, i.e. when two categories possess same local attributes and differ only in the way the attributes combine to generate the global view of the object, relation-agnostic approaches do not capture the full semantic information in an input image. This hypothesis is then proved by theory and validated in the experimental parts, which is the main theoretical contribution of the paper. I do like this novel perspective.

Weakness:
I do not have major concerns regarding the technical details, however, the visualization analysis of the proposed method is lacking, for example, under what circumstance that the proposed model can significant improves the performances and when does it fail.

---

> ### Author Response · Authors · 2022-08-02
> **Response to Reviewer G6W7**
>
> We thank the reviewer for suggesting this qualitative evaluation, as it has allowed us to illustrate our model in a more transparent manner, as well as better understand the situations that might limit its potential. We have now added qualitative classification results, and depictions of the predicted cross-view relationships in Figure 4 and Section 2.3 Qualitative Classification Results of the Supplementary Material.
>
> **How we obtained visual representations for the cross-view relationships:** The cross-view relationships are depicted in Figure 4 via a graph of the local views. The graph represents the manner in which the local views combine to form the overall object. The nodes of the graph represent the local views. The nodes are connected based on the mutual attention scores of their corresponding representations obtained from the final layer of the Attribute Summarization Transformer (AST). The weight of the edge is proportional to the magnitude of attention. For the purpose of simplicity, we depict fewer local views in the visualization, than are actually used for computation.
>
> **Observations:** It can be seen that images that provide a diverse set of local views, and thus, a larger space of possible cross-view relationships are the ones that get classified correctly with full certainty. However, as the number of unique local views get limited (possibly due to occlusion or an incomplete photographing of the object), it reduces the amount of relational information that can be mined. Under situations when even the individual local-views are largely shared between classes, there remains no discriminative premise (neither local/global, nor relational) for telling their instances (with limited depiction of local views) apart. It is under such circumstances that the classifier gets confused.
>
> **Example:** For instance, in the example from the CUB dataset (the top row in Figure 4), the images of the Acadian Flycatcher and Bank Swallow depict sufficient numbers of local views like the head, tail, belly and wings, which provide a large space of potential cross-view relationships that favor classification outcome. On the other hand, the images of the Black-footed Albatross and Laysan Albatross only depict the head and the neck, thus limiting the number of computable relationships that can act as discriminators. Moreover, the head and the neck look largely similar between the two categories, thereby leading to cross-category confusion causing a subsequent misclassification. However, we believe that such a situation can be addressed by learning different distributional priors over the set of local views, which we plan to take up as future work.
>
> We will include these findings in the final version of the main manuscript.

---

### Official Review · Reviewer_EFjT · 2022-07-13

**Rating:** 7
**Confidence:** 3
**Soundness:** 3 good
**Presentation:** 3 good
**Contribution:** 3 good

**Summary:**

In the fine-grained setting, discriminating between different classes requires learning how different local parts combine to form the object of interest. In this work, the authors introduce a theoretical framework and a novel method that decomposes FGVC tasks into relation-agnostic feature extraction and cross-view relation learning. They show the superiority of such method through a set of experiments.

**Questions:**

1. In Definition 1, k-distinguishability is defined with respect to two classes only. What does this imply at the level of the entire dataset. What happens if different pairs of classes have different corresponding k values?
2. In settings where the datasets includes multiple subgroups of classes (cats, dogs, birds) that would be coarsely separable at the group level,  but would require fine-grained visual modeling within each group (in birds: white-faced plover vs. kentish plover). How would this approach change? How is k-distinguishability defined? How is the number of relational proxies c selected?


**Limitations:**

The authors discuss one of the main limitation, being the local view generation.


**Strengths And Weaknesses:**

The problem this work address is relevant. While I am unable to gauge the relevance of the proposed theoretical framework and its broad usefulness to the community, the proposed approach is solid and of interest.

Strengths:
1. **The experimental setup is solid.** The authors test their method on multiple datasets and consistently show competitive results across all of them. These results support the idea of modeling the cross-view relation in their proposed architecture. This is further demonstrated through ablation studies, that highlight the need for a cross-view relational function and provide further insights into the method.
2. The theoretical framework established in this work is clear, and while it builds on a lot of definitions, the proofs are simple and easy to follow.

Weaknesses:
1. **The permutation invariance property is counter-intuitive** The authors introduce the permutation invariance property as a necessary property for a model to solve FGVC tasks. In section 3.4, they introduce a novel transformer (AST) that does not use positional embeddings and is thus permutation invariant. While the permutation invariance property can provide certain desired properties like potentially better generalization, the authors do not provide theoretical evidence for why it would be necessary, and it is not fully clear how it is motivated either. Certain claims in the introduction would actually suggest otherwise: “differ only in the way the attributes combine to generate the global view of the object” or “[features like] the distance between the head and the body, or the angular orientation of the legs”. This suggest that features like positional encoding would actually be critical.
2. The term “relation” is not explicitly defined and it is unclear what the authors mean. The "relation-agnostic representation” is established in Definition 4, and while it is clear what it means in mathematical terms, its relation to FGVC problems is not evident. Providing more clarifications would make the text easier to follow.
3. **Decomposing the problem into relation-agnostic encoder and a cross-view relational function.** The authors do not argue for why this decomposition is necessary when solving FGVC problems- at least in the main text. A discussion of this can be found in appendix 6.3, and I would argue for including this in the main text as it better explains the idea of the relational gap and seems to at least provide an initial motivation for this decomposition.

---

> ### Author Response · Authors · 2022-08-02
> **Response to Reviewer EFjT - Part 2 of 2**
>
> 3. **Decomposing the problem into relation-agnostic encoder and a cross-view relational function:** Similar to the reason mentioned in the answer for Question 2, factorizing the label information in terms of cross-view relationship provides us with a clean framework for analyzing existing literature, precisely identifying the gaps therein, and thus, ways of resolving the same. Identity 1 in Appendix 6.1 proves that given a relation-agnostic encoder (any SotA encoder), the only uncertainty that remains in its representation space stems from the cross-view relational information. Thereafter in Proposition 2, we prove that given a relation-agnostic encoder, there needs to be a distinct sub-model for learning the cross-view relational information in order for a learner to qualify as being sufficient, thus requiring the said problem decomposition.
> We will incorporate this, along with the discussion in Appendix 6.3 in the main text following Proposition 2.
>
> 4. **Value of $k$ for multi-class datasets:** If a multi-class dataset contains class-pairs with differing $k$-values, in our current implementation, we consider a single $k$-value for the entire dataset by choosing the largest $k$ considering all class-pairs in that dataset. Formally, the $k$ value for the entire dataset is given by $max [ k_{ij}; \forall (i, j) ]$, where $i$ and $j$ are class indices.
> This is a theoretically valid choice since $k$-distinguishability puts a lower bound on the number of local views, and thus, any value higher than the true $k$ should also work. We empirically validate that this is a functionally correct design via our ablation in Figure 2. The performance can be seen to saturate beyond the maximum $k$ value for the entire dataset.
> We agree that this is not the choice that provides the most computational efficiency, as for many class pairs, the actual value of $k$ would be lower than the global value for the entire dataset, leading to unnecessary computations over redundant views. This however, is a problem that deserves to be researched on its own. In fact, this is the exact direction we are pursuing as a follow-up to this work via generating explanations for the relational embeddings and pruning out local-views that do not feature in that explanation. So we are glad the reviewer brought this up, as this gives us the approval that this is a sensible next-step to take.
>
> 5. **$k$-distinguishability for coarse-grained categories:** For the coarse-grained categories, we need not resort to the idea of $k$-distinguishability, as because of the large inter-class differences, most of them would be separable just via their global views (as has been demonstrated by most SotA classifiers on coarse-grained datasets like ImageNet).\
> **Choice of $k$:** The choice of $k$ in such a setting would be analogous to the purely fine-grained case, the maximum $k$ value among all the leaf-level / fine-grained class-pairs. The only drawback being a lot of redundant computations to tell apart classes that come from different coarse-grained categories.\
> **Number of relational proxies:** The number of relational proxies $c$ could be chosen to be the total number of fine-grained classes across all coarse-grained categories. The proxies could be grouped based on their corresponding higher level super-category in the dataset. The coarse grained class of an image would then be the super-category that its fine-grained proxy belongs to.

---

> > ### Comment · Reviewer_EFjT · 2022-08-08
> > **Response to rebuttal**
> >
> > I thank the reviewers for their response, and appreciate their efforts in providing additional clarifications and revising the paper. The analogy provided in response 1 was intuitive and helped me better grasp the motivation behind this framework, the additional ablation that was performed experimentally supports this intuition.
> >
> > Overall, all the clarifications that were provided help with the understanding of this work, and I am happy to see the paper revised accordingly.
> >
> > Many of my concerns about clarity of the method as well as its limitations have been addressed, and I have, therefore, increased my score from a 6 to a 7.

---

> > > ### Author Response · Authors · 2022-08-09
> > > **Note of thanks**
> > >
> > > We thank the reviewer for taking the time to go through the rebuttal, appreciating our intuitive explanations and additional experiments, and increasing their score.

---

> ### Author Response · Authors · 2022-08-02
> **Response to Reviewer EFjT - Part 1 of 2**
>
> 1. **Necessity for Permutation Invariance:** We thank the reviewer for pointing this out. We agree that we have only rather glossed over the property of permutation invariance, and its necessity in constructing the relationship modelling function ($\xi$, line: 108) is not entirely apparent. As the reviewer has correctly observed, an equivalent formulation is possible by replacing the AST and the view-unifier MLP ($\rho$) with a transformer having position embeddings. However, we observed that this setting led to accuracies that were 0.2-0.3% lower on FGVC Aircraft and Stanford Cars than our current approach. The reason this happens is because there are two underlying sub-problems to solve, and tasking a transformer with doing it end-to-end leads to highly entangled intermediate representations that make learning the relationships challenging. We provide more details on the sub-problems below. We instead train our AST to solve only one of the sub-problems, and our view-unification MLP the other, thus achieving the same functionality, but in a factorized manner that aids convergence.\
> **Intuitive Analogy:** The problem of local-to-global relation computation can be viewed as a bit-string-to-integer matching problem.
> Consider 3 bits, say $b_1, b_2$ and $b_3$, corresponding to 3 local views. Let the global view be represented by an integer that can be encoded with 3 bits, say with a value of $g = 6$, for this example. The problem then is to find the association of the integer 6 with its corresponding binary representation 110. This association represents the cross-view relationship.
> Drawing a parallel with our algorithm, the first step towards solving this problem is to enumerate all the possible ways in which the local views can combine (to produce any global view, not specifically $g$), given by $S = [000, 001, 010, ….110, 111]$. The bit values encode the presence or absence of a particular view in the cross-view relationship. So, no matter what order we observe $b_1, b_2$ and $b_3$ in, we must output the same set $S$, as it is required to be an exhaustive enumeration. This is exactly what the AST achieves.
> Once we have S, the next step is to design a function that finds the mapping $S, g \mapsto 110$, i.e, the correct binary encoding for the integer $g = 6$, which is accomplished by $\rho$ in our method.\
> **Purpose:** As illustrated through the above analogy, one can view the local-to-global relationship modelling function as an enumerative search algorithm - given a set of local views, it first *enumerates* all possible ways in which they can combine to form a meaningful global view. Given that enumeration, it then *finds* the target solution by learning to identify the correct combination that matches with the global-view representation. Thus, the *enumerate* operation needs to be permutation invariant, as it has to consider all possible combinations of the inputs, and the *find* operation needs to be a view-unifier by construction.\
> **Motivation:** Behind our specific design choice was the motivation to keep the *enumerate* and *find* steps separate. This allows the model to have dedicated representation spaces for the two distinct subtasks, which in turn facilitates better convergence. Our AST thus produces the candidate *enumerations* of local-view aggregations, and the view-unification MLP ($\rho$) *finds* the correct aggregation that matches with the global view.
> In the final version of our paper, we will incorporate the above intuition, purpose and motivation alongside the definition of and requirement for permutation invariance.
>
> 2. **Relation:** Intuitively, what we mean by “relation” here is the way the local parts of an object combine to form its global view (lines 18 - 19).  In Section 1, Introduction of the main paper, we provide an intuitive example to illustrate this (lines 21 - 26). Mathematically, it is what manifests as the Information Gap (Section 3.2, Proposition 1) in a relation-agnostic representation space, quantified as $I(\mathrm{\mathbf{x}}; \mathrm{\mathbf{r}} | \mathrm{\mathbf{z}}) = I(\mathrm{\mathbf{x}}; \mathrm{\mathbf{y}}) - I(\mathrm{\mathbf{z}}; \mathrm{\mathbf{y}})$.\
> **Relation-agnostic representation:** Relation-agnostic representations are ones that are obtained by independently processing all the views, without taking into account the cross-view relational information (as described above). All existing FGVC works can be categorized under this head (sharing the primary objective of identifying discriminative object parts in an isolated manner). We introduced the idea of relation-agnostic representations with the goal of formalizing this commonality across the existing literature.

---

### Official Review · Reviewer_9xUV · 2022-07-14

**Rating:** 5
**Confidence:** 4
**Soundness:** 3 good
**Presentation:** 3 good
**Contribution:** 3 good

**Summary:**

Summary.

This paper is dedicated to developing algorithms for fine-grained image recognition. They argue it is not enough to distinguish fine-grained categories only based on partial information. Therefore, they propose relational proxies, which leverage the relational information between the global and local views of an object. They also provide theoretical explanations to support the effectiveness of the proposed methods. Experiments on six fine-grained benchmark datasets offer positive results.

**Questions:**

Refer to the weakness section.

**Limitations:**

Both limitations and potential negative social impacts are discussed in the submission.

**Strengths And Weaknesses:**

Pros.

1. The proposed methods make sense and are well-motivated. Both theoretical and empirical analyses are provided to support the effectiveness.

2. The paper is well written and easy to follow. Figure 1 is informative and illustrative.



Cons.

1. There are missing numbers in Table 1. For a comprehensive comparison, it is necessary to complete it. Minor: the caption of tables should be on the above content.

2. The performance gains are marginal, especially on CUB (0.3%) and NA Birds (0.2%)? Any explanations? It seems the proposed methods are less working for bird images. Meanwhile, the error bar is required for Table1 since the current accuracy margin is too small.

3. More network backbones are needed to support the generalization of proposed methods across architectures.

4. As for the study in Figure 2, the number of local views |L| and the size of the local patch should be correlated. A detailed analysis is needed.

---

> ### Author Response · Authors · 2022-08-02
> **Response to Reviewer 9xUV**
>
> 1. **Missing numbers in Table 1:** Initially, only considering the numbers reported in the original publications, a larger fraction of the accuracy scores for the SotA methods in Table 1 were missing. We tried our best to run their implementations and fill-out as many of the blanks as we could. For the ones that remain vacant, it is either because we were unable to find existing implementations of their method (MaxEnt [11]), or it was difficult to get it running even if one was available (DBTNet [50] and CAP [3]).
> 2. **Error Bars:** We thank the reviewer for this suggestion. We will add the error bars (over 5 independent runs with different random seed initializations) for Table 1 in the final version of the paper. For completeness, we also present them below:
>
> |Dataset | Accuracy (mean ± std) |
> | :- | :-: |
> |FGVC Aircraft  | 95.25 ± 0.02 |
> |Stanford Cars | 96.30 ± 0.04 |
> |CUB | 92.00 ± 0.01 |
> |NA Birds| 91.20 ± 0.02 |
> |Cotton| 69.81 ± 0.04 |
> |Soy| 51.20 ± 0.02 |
>
> It can be seen that our method provides stable performance across different initializations, with a highest error bound of only ± 0.04% across all 6 datasets.\
> **Marginal Performance Gains:** FGVC being a highly challenging problem domain, we observe that most recent SotA are only able to improve marginally over their predecessors. For instance, the differences between TransFG [13] and CAP [3] are just as low (0.2%) for NA Birds or even lower for CUB (0.1%). Among all fine-grained datasets, the Bird datasets are relatively more difficult to categorize due to challenges like shift in the distribution of bird poses in test-set images [4], occlusions, and high intra-class and low inter-class variations. Thus, the SoTA performance on Birds datasets is somewhat lower compared to other datasets, despite having been around for a long time. We understand that these marginal performance gains on the long-studied datasets may not be fully convincing of the efficacy of our method. For that reason, we also considered the Cotton and Soy Cultivar datasets [41]. They have been newly proposed in 2021 and provide a highly challenging novel setting with extremely low inter-class variations for FGVC algorithms to address. We show consistent and stable performance gains of over 4% on both these datasets (Table 1).
>
> 3. **Network backbone:** We have now evaluated our model with the VGG-16 backbone on the FGVC Aircraft and CUB datasets. We present our results below with error bars, and compare them to existing SotA methods that also provide their accuracies with VGG-16 backbone.
>
> | |FGVC Aircraft|CUB
> |-|:-:|:-:|
> MaxEnt [5]|78.08|77.02
> MMAL [6]|87.00|83.75
> Ours|**91.20** ± 0.03|**88.13** ± 0.01
>
> As the numbers show, our method remains stable across backbones, significantly outperforming SotA methods with VGG-16 backbones as well.
>
> 4. **Correlation between number of local views and size of local patches:** We sincerely thank the reviewer for suggesting this experiment as we believe that doing such a correlation study is a great way of determining the right computational trade-offs for our method. We trained our model on FGVC Aircraft by varying the number of local views |L| and the size of each local patch to identify their correlations. We present our results in the table below, where rows represent the number of local views |L| and the columns represent the side-length of each local patch. So, if the global view has spatial dimensions size x size, each local patch would be of size/t x size/t, where t is the scaling factor that is varied across the columns. In summary, the rows represent increasing the number of local views top-down, and the columns represent increasing the patch size left-to-right.
> The numbers are expressed as relative deviations from a reference of 95.25%, i.e., the setting corresponding to Row 1, Column 3, whose performance we reported in Table 1 of the main paper.
>
> ||size/5|size/4|size/3|size/2
> |:-:|:-:|:-:|:-:|:-:|
> **7**|-0.03|-0.02|0.00|-0.14
> **12**|+0.01|+0.02|0.00|-0.11
> **15**|+0.05|+0.03|+0.01|-0.11
> **18**|+0.05|+0.03|+0.00|-0.10
>
> From the table above, we can see that increasing the patch size beyond a certain point has a  detrimental effect, as with increasing size, the local views tend to lose their granularity and degenerate into global views. Increasing the number of crops has a stronger improvement effect on performance if the patch size is small. However, decreasing the patch size at the cost of an increased number of local views also has its downsides - the number of attention computations in the attribute summarization step increases quadratically. Thus |L| and local patch size need to be determined based on application specific accuracy requirements and the amount of available computation resources.
> We will include the results of this experiment and our observations in the final version of the paper.

---

### Official Review · Reviewer_1ZBu · 2022-07-17

**Rating:** 5
**Confidence:** 3
**Soundness:** 2 fair
**Presentation:** 2 fair
**Contribution:** 2 fair

**Summary:**

The authors propose Relational Proxies, a novel approach that leverages the relational information between the global and local views of an object for encoding its semantic label.

**Questions:**

Insufficient experiments and limited performance gains.

**Ethics Review Area:**

["I don’t know"]

**Limitations:**

N/A.

**Strengths And Weaknesses:**

I think the main novelty comes from the introduced Relational Proxies and the corresponding comprehensive theoretical and experimental analysis.

Weaknesses One area of improvement for the paper at hand would be clarity, especially with respect to the exposition of the proposed architecture. It takes multiple read throughs in order to identify the actual architecture proposed.

Lack of experiments on larger datasets. In a time of ever-growing dataset sizes it would be good to provide and compare results of suchmodels when trained on larger datasets. This is important for judging the impact asimprovements stemming from architecture engineering typically vanish with growing dataset sizes.

---

> ### Author Response · Authors · 2022-08-02
> **Response to Reviewer 1ZBu**
>
> 1. **Clarity on model architecture:** We thank the reviewer for this suggestion. In Section 3.4, we will provide a dedicated summary of the entire architecture that gives a simple and clear overview of the full model end-to-end, and directly correlate it with the components depicted in the model diagram (Figure 1).
> 2. **Experiments on large datasets:** We follow recent state-of-the-art literature to choose datasets for evaluation [3,11,13,41,47,50]. However, we agree that it is important to evaluate the scalability of our method by considering large-scale datasets. For that purpose, we are currently training and evaluating our model on the iNaturalist 2017 dataset [A], which has a total of 5,089 categories, 675,170 train + val images, and 182,707 test images. We have chosen iNaturalist because (1) it contains large number of fine-grained categories from diverse super classes including Plant, Insect, Bird, Mammal, and so on; (2) it is highly imbalanced with very different number of images per category, which we believe can stress test our proposed model for appropriate validation. Nevertheless, we would also be happy to evaluate our model on any other datasets that the reviewer would like to suggest. We will update this rebuttal via a comment, and also the final version of the paper, with the corresponding results on iNaturalist once they are available.
> 3. **Limited performance gains:** Below, we present the error bars (over 5 independent runs with different random seed initializations), which we will also add to Table 1 in the final version of the paper:
>
> |Dataset | Accuracy (mean ± std) |
> | :- | :-: |
> |FGVC Aircraft  | 95.25 ± 0.02 |
> |Stanford Cars | 96.30 ± 0.04 |
> |CUB | 92.00 ± 0.01 |
> |NA Birds| 91.20 ± 0.02 |
> |Cotton| 69.81 ± 0.04 |
> |Soy| 51.20 ± 0.02 |
>
> It can be seen that, although with narrow gains, our method provides stable performance across different initializations, with a highest error bound of only ± 0.04% across all 6 datasets.
>
> FGVC being a highly challenging problem domain, we observe that most recent SotA are only able to improve marginally over their predecessors. For instance, the differences between MMAL [47] and CAP[3] on FGVC Aircraft, TransFG [13] and CAP [3] on CUB and NA Birds are even lower than us, or just as low. We understand that these marginal performance gains on the long-studied datasets may not be fully convincing of the efficacy of our method. For that reason, we also considered the Cotton and Soy Cultivar datasets [41]. They have been newly proposed in 2021 and provide a highly challenging novel setting with extremely low inter-class variations for FGVC algorithms to address. We show consistent and stable performance gains of over 4% on both these datasets (Table 1).
>
> **Additional References:** \
> [A] Grant Van Horn, Oisin Mac Aodha, Yang Song, Yin Cui, Chen Sun, Alex Shepard, Hartwig Adam, Pietro Perona, Serge Belongie. The iNaturalist Species Classification and Detection Dataset. *In* CVPR, 2018.

---

> > ### Author Response · Authors · 2022-08-04
> > **Evaluation on a large scale dataset - iNaturalist 2017**
> >
> > We have now completed our experiment on the iNaturalist 2017 dataset. The results are presented below:
> >
> > ||iNaturalist 2017
> > |:-|:-:|
> > TransFG [13] |71.70
> > **Ours (Relational Proxy)**|**72.15**
> >
> > It can be seen that our method provides an improvement of 0.45% over TransFG [13], which is the current state-of-the-art on iNaturalist. This shows that our method does in fact scale to very large datasets, supporting the generality of our theory and design choices.
> > We are currently in the process of performing additional training runs with different random seed initializations on INaturalist to obtain its error bounds. However, as results from all other datasets show, our method offers very stable results with extremely low standard deviation. In the final version of the paper, we will add the mean accuracy along with the standard deviation obtained from 5 independent runs with different initializations.

---

### Comment · Area_Chair_FMA1 · 2022-08-06
**Please reply to rebuttals**

Dear Reviewers,

Thanks for your work reviewing this paper. There are only a few days left for discussing with the authors.

Please read the authors' rebuttals and **explain why you decided to keep your score as is or why you updated it.** (not just click on the "acknowledge" button).  It is very frustrating for authors to be completely ignored.

Hence, we urge you to read and reply ASAP.

Thanks again,

AC

---

### Meta-Review · Area_Chair_FMA1 · 2022-08-26

**Recommendation:** Accept
**Confidence:** Certain

**Metareview:**

This paper proposes a novel approach for fine-grained image recognition, which utilizes the relational information between the global and local views of an object.  It is a reasonable and important finding that not only representing local parts but relating them are critical to establishing superior performance.  The authors validate their proposal’s effectiveness with both theoretical explanations and positive empirical results on various benchmarks.  The authors also did a great job in rebuttal. They provide more clarifications, extra experiments on large datasets, and newly included error bars.    Most of the reviewers are satisfied with the rebuttals and discussions,  and all reviewers have a consistent recommendation.  We think this paper can bring new insights to the visual recognition community and help people understand how the key features and their relations work.   Please also include the newly added experiments and clarifications in the new revision.


**Award:**

No

---

### Decision · Program_Chairs · 2022-09-14

Accept